 

# Increase of circulating IGFBP-4 following genotoxic stress and its implication for senescence

Nicola Alessio[1], Tiziana Squillaro[1], Giovanni Di Bernardo[2], Giovanni Galano[2], Roberto De Rosa[2], Mariarosa AB Melone[3], Gianfranco Peluso[4], Umberto Galderisi[1,4,5]*

[1]Department of Experimental Medicine, Biotechnology and Molecular Biology Section, University of Campania "Luigi Vanvitelli,", Naples, Italy; [2]ASL Napoli 1 Centro P.S.I. Napoli Est - Barra, Naples, Italy; [3]Department of Medical, Surgical, Neurological, Metabolic Sciences, and Aging, 2nd Division of Neurology, Center for Rare Diseases and InterUniversity Center for Research in Neurosciences, University of Campania 'Luigi Vanvitelli', Naples, Italy; [4]Research Institute of Terrestrial Ecosystems (IRET), CNR, Naples, Italy; [5]Sbarro Institute for Cancer Research and Molecular Medicine, Center for Biotechnology, Temple University, Philadelphia, United States

**Abstract** Senescent cells secrete several molecules, collectively named *senescence-associated secretory phenotype* (SASP). In the SASP of cells that became senescent following several in vitro chemical and physical stress, we identified the IGFBP-4 protein that can be considered a general stress mediator. This factor appeared to play a key role in senescence-paracrine signaling. We provided evidences showing that genotoxic injury, such as low dose irradiation, may promote an IGFBP-4 release in bloodstream both in mice irradiated with 100 mGy X-ray and in human subjects that received Computer Tomography. Increased level of circulating IGFBP-4 may be responsible of pro-aging effect. We found a significant increase of senescent cells in the lungs, heart, and kidneys of mice that were intraperitoneally injected with IGFBP-4 twice a week for two months. We then analyzed how genotoxic stressors may promote the release of IGFBP-4 and the molecular pathways associated with the induction of senescence by this protein.

*For correspondence:
umberto.galderisi@unicampania.it

Competing interests: The authors declare that no competing interests exist.

## Introduction

Senescence is a process that occurs following genotoxic stimuli and induces permanent cell cycle arrest with a loss of cellular functions. Senescence is a complex phenomenon that may produce different biological outcomes. There is evidence correlating senescence to aging, since it contributes to the reduction of tissue functions, cell renewal, and homeostasis (*Campisi, 2013*; *He and Sharpless, 2017*). Senescence also has a contrasting effect on cancer: in some conditions, it represents a protective anticancer mechanism leading to the arrest of transformed/damaged cells; in other settings, senescent cells may promote cancer growth. In recent years, new findings have shown a role for senescence during embryonic development and in tissue recovery after wound injuries (*Campisi, 2013*; *Campisi and d'Adda di Fagagna, 2007*). A cell that becomes senescent loses its original function and acquires new activities. This is mainly accomplished by the secretion of several molecules, for which the term *senescence-associated secretory phenotype* (SASP) has been proposed (*Coppé et al., 2010*). SASP contains growth factors, inflammatory cytokines, chemokines, and other bioactive molecules. It represents a danger signal and sensitizes normal neighboring cells to senesce (paracrine activity) and reinforces the senescence process through autocrine signaling. SASP factors

induce tissue remodeling and immune cell recruitment (*Coppé et al., 2010*). Autocrine and paracrine activity of SASP is well documented, while less is known about possible SASP long-distance effect. Recently, some studies evidenced that SASP may induce long-distance bystander effects both on normal and cancer cells (*Borodkina et al., 2018*; *Gonzales-Puertos et al., 2015*).

On this premise, we hypothesized that senescent cells close to circulatory flow may release SASP components into bloodstream and hence pro-inflammatory and pro-aging factors may reach organs and tissues that are quite distant from the site of SASP production. We focused our attention on IGFBP proteins that are released by several senescent cell types, such as endothelial and epithelial cells, fibroblasts and mesenchymal stromal cells (*Baxter, 2014*; *Gonzales-Puertos et al., 2015*; *Severino et al., 2013*). IGFBP proteins modulate the function of IGF-I and IGF-II, which are growth factors involved in the regulation of the growth, survival, and differentiation of several cell types (*Baxter, 2014*; *Mohan and Baylink, 2002*). In our previous studies, we demonstrated that SASP produced by replicative senescent mesenchymal stromal cells (MSCs) contained the protein IGFBP-4, a member of IGFBP family, which appeared to play a key role in senescence-paracrine signaling, since its inactivation greatly reduced the pro-senescence activities present in SASP (*Severino et al., 2013*). Indeed, healthy MSCs underwent senescence when incubated with SASPs produced by replicative senescent MSCs; this SASP property is lost when IGFBP-4 is blocked with neutralizing antibodies. Moreover, the addition of recombinant IGFBP-4 to MSC cultures triggers senescence (*Severino et al., 2013*). In the current study, we analyzed how genotoxic stressors may promote the release of IGFBP-4 and the molecular pathways associated with the induction of senescence by this protein.

## Results

### IGFBP-4 is a general stress mediator and is released following different genotoxic injuries

Induction of chronic senescence occurs after periods of progressive stress, such as that associated with DNA replication (*van Deursen, 2014*). In a previous research, we evidenced that the replicative (chronic) senescence of MSCs is associated with IGFBP-4 release. Here we showed that IGFBP-4 is also secreted following induction of acute senescence, defined as acute increase in specific stress (*van Deursen, 2014*). MSCs treated with doxorubicin, peroxide hydrogen ($H_2O_2$), and low- and high-dose X rays became senescent and released IGFBP-4 (*Figure 1—figure supplement 1a,b,c*). This and our previous results evidenced that different stressors, either in acute or chronic conditions, promote the release of IGFBP-4 in the senescent cell secretome (*Severino et al., 2013*). We then performed a follow-up analysis to evaluate the timing of the IGFBP-4 release after stress induction. In $H_2O_2$ treated cells, we detected an increase in IGFBP-4 secretion 24 hr poststress; this was further augmented at 48 hr, and then it slightly declined (*Figure 1—figure supplement 1d*). This suggests that IGFBP-4 is released during the same time period of the senescence onset and further indicates that it may have a role in this process.

### In vivo experiments suggest a possible role for IGFBP-4 as a stress signal

We evaluated whether genotoxic injury may promote an IGFBP-4 release in vivo, having demonstrated in vitro that senescence induced by stressful insults is associated with IGFBP-4 secretion. We selected low dose irradiation as genotoxic stressor since there are several findings showing that X and gamma rays at low dose can induce cellular senescence in several cell types (*Alessio et al., 2017b*; *Squillaro et al., 2018*). Moreover, studies on the biological effects of low dose radiations are of great interest for human health. We irradiated five mice (C57BL/6 strain) with a low dose of X ray (100 mGy) and collected sera 6, 24, 48, 72, and 144 hr postirradiation. At all of the analyzed time points, we observed a statistically significant increase ($p<0.05$) of IGFBP-4 compared with five untreated animals (*Figure 1a*). We collected several organs 48 and 144 hr postirradiation and evidenced the presence of senescent cells in lung, heart and kidney (*Figure 1b*). We also isolated MSCs from mice following irradiation and observed increase senescence compared with the controls (*Figure 1c*). MSC senescence, we detected with acid beta galactosidase assay, was confirmed by evaluating the expression of several proteins associated with this phenomenon (*Figure 1e*).

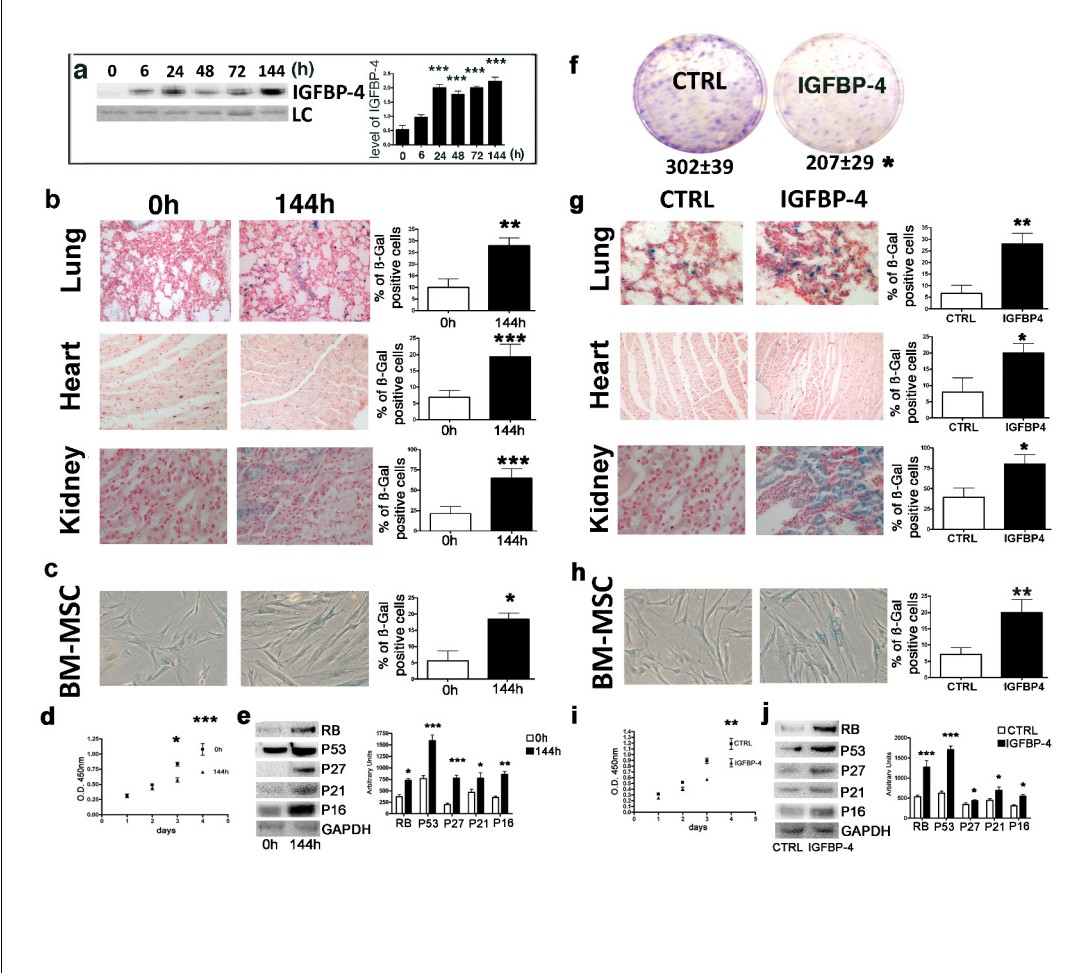

**Figure 1.** Genotoxic stress and IGFBP-4. Panel a - Western blot analysis of IGFBP-4 in the sera of mice treated with 100 mGy X ray. The sera were collected at different time points following irradiation. Membrane staining with Ponceau S acid red was used as a loading control (LC). The graph shows a representative densitometric analysis of the IGFBP-4. We compared the untreated mice (0 hr) with all of the other experimental conditions, and the statistical differences are indicated with ***p<0.01. The data are expressed in arbitrary units. For each time point, we used three irradiated mice and three controls. Panel b - In situ acid beta galactosidase analysis on the histological sections obtained from mice 144 hr after X ray irradiation. The picture shows representative sections of the lungs, hearts and kidneys. The senescent cells are shown in blue. We used three irradiated mice and three controls. The graphs represent the percentage of beta-galactosidase-positive cells determined on at least five histological sections. The statistical differences are indicated with the ** (p<0.01) and *** (p<0.001) symbols. For experimental group we used three mice. Panel c -The picture shows a representative image of the acid-beta galactosidase assay on the bone marrow MSCs (passage 1) obtained from the control and irradiated animals (144 hr post treatment). We used three irradiated mice and three controls. The statistical difference is indicated with the * (p<0.05) symbol. Panel d – Cell proliferation of bone marrow MSCs (passage 1) was determined by Cell Counting Kit-8 (CCK-8) colorimetric assay. The graph shows data coming from control and irradiated animals. The symbols * and *** denote p<0.05 and p<0.001, respectively. Panel e – Western blot analysis of senescence-related proteins in bone marrow MSCs (passage 1) obtained from irradiated and control animals. GAPDH expression was used as the loading control. The graph shows the densitometric analysis and the statistical differences are indicated with *p<0.05 or **p<0.01 or ***p<0.001. The data are expressed in arbitrary units. Panel f – The CFU assay performed on the bone marrow MSCs (passage 1) obtained from the control and the IGFBP-4-treated animals. The number of CFU clones per 1000 plated cells is reported. The statistical difference (p<0.05) is indicated with the * symbol. For experimental group we used three mice. Panel g – In situ acid beta galactosidase analysis on the histological sections obtained from mice after two months of treatment with IGFBP-4 (intraperitoneally injected twice per week). The picture shows representative sections of the lungs, heart, and kidneys in the control and the IGFBP-4 treated animals. The senescent cells are shown in blue. The graphs represent the percentage of beta-galactosidase-positive cells determined on at least five histological sections. The statistical differences are indicated with the * (p<0.05) and ** (p<0.01) symbols. For experimental group we used three mice. Panel h -The picture shows a representative image of the acid-beta galactosidase assay on the bone marrow MSCs (passage 1) obtained from the control and IGFBP-4 treated animals. For experimental group we used three mice. Panel i – Cell proliferation of bone marrow MSCs (passage 1) was determined by Cell Counting Kit-8 (CCK-8) colorimetric assay. The graph shows data coming from control and IGFBP-4 treated animals. The symbols ** indicates p<0.05. Panel j – Western blot analysis of senescence-related proteins in bone marrow MSCs (passage 1)

*Figure 1 continued on next page*

*Figure 1 continued*

obtained from control and IGFBP-4 treated animals. GAPDH expression was used as the loading control. The graph shows the densitometric analysis and the statistical differences are indicated with *p<0.05 or ***p<0.001. The data are expressed in arbitrary units.

The online version of this article includes the following source data and figure supplement(s) for figure 1:

**Source data 1.** Original image data (gels and micrographs) for *Figure 1*.
**Source data 2.** Numerical data for *Figure 1*.
**Figure supplement 1.** Senescence induced by exogenous stressors and IGFBP-4 release.
**Figure supplement 1—source data 1.** Original image data (gels and micrographs) for *Figure 1—figure supplement 1*.
**Figure supplement 1—source data 2.** Numerical data for *Figure 1—figure supplement 1*.

Moreover, we observed a decline in cell proliferation (*Figure 1d*) along with a decrease of clonogenic potential (360 ± 36 clones per 1000 plated cells in control vs 290 ± 23 clones in irradiated mice). This preliminary experiment showed us that a genotoxic insult may induce cellular senescence along with an increase of IGFBP-4 in serum. We then questioned if an increased level of circulating IGFBP-4 may be responsible of pro-aging effect. We intraperitoneally injected mice with 1 µg IGFBP-4 twice a week for two months. This treatment produced an increase of IGFBP-4 serum levels. We then evaluated whether this increase in IGFBP-4 might be associated with the senescence phenomena. We found a significant increase of senescent cells in the lungs, heart, and kidneys (*Figure 1g*). We also isolated MSCs from the IGFBP-4-treated animals and the control animals. MSCs from the IGFBP-4 group showed a significant increase in senescence compared with the controls (*Figure 1h*), and this was related to impaired proliferation (*Figure 1i*) and reduced stemness, as demonstrated by the CFU assay (*Figure 1f*). Senescence was confirmed by detecting the expression of some proteins associated with this phenomenon (*Figure 1j*).

## IGFBP-4 is a stress mediator also in humans

Biological and molecular functions in humans can be deduced by using the mouse as animal model. However, several findings reveal a divergence between molecular signaling pathways among species

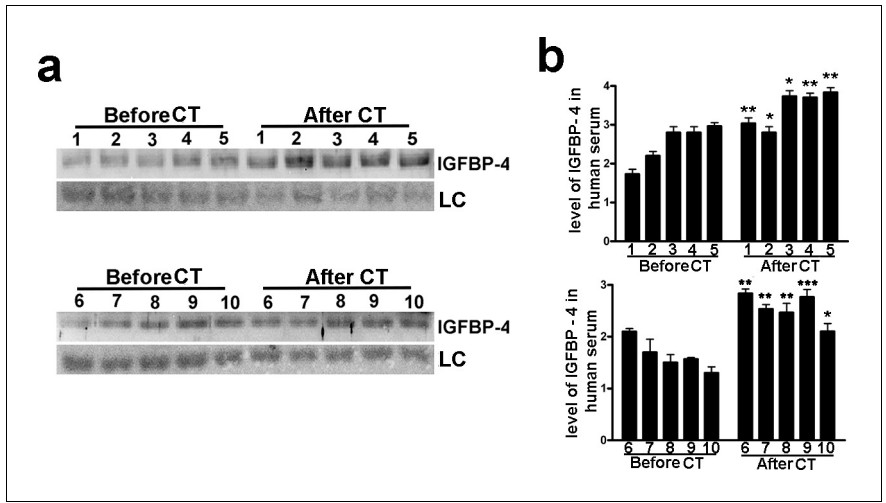

**Figure 2.** Low dose of X ray induced increase in serum level of IGFBP-4. The picture shows the western blot analysis of IGFBP-4 in the sera of 10 patients before and 48 hr after abdominal CT scan. Membrane staining with Ponceau S acid red was used loading control (LC). The graph shows the densitometric analysis of the IGFBP-4 level. Each patient is indicated with a number. For each patient the data are expressed as arbitrary units ( ± SD, n = 3 technical replicates) and the significant difference between samples harvested before and after CT is indicated with ** (p<0.01) or *** (p<0.001).

The online version of this article includes the following source data for figure 2:

**Source data 1.** Original image data (gels and micrographs) for *Figure 2*.
**Source data 2.** Numerical data for *Figure 2*.

as close as mammals such as mice and humans. We evidenced that molecular pathways governing senescence process in mouse and humans do not overlap completely (*Alessio et al., 2017a*). For this reason, we aimed to evaluate if genotoxic stress may induce increase of circulating IGFBP-4 in humans. We determined the IGFBP-4 levels in sera of 10 patients that received abdominal CT scan. The choice was based on the consideration that this is among the CT analyses with the highest effective dose of radiation per scan (*Schegerer et al., 2017*). We collected patients' sera before CT scan and 48 hr later and evidenced a significant increase of IGFBP-4 levels following irradiation treatment (*Figure 2*).

### What are the signaling pathways associated with IGFBP-4 release and its pro-aging activity?

Having demonstrated an association between genotoxic stress and IGFBP-4, we aimed to address some key questions to dissect the role of IGFBP-4 in senescence. What is the signaling pathway that promote the IGFBP-4 release following the stress phenomena? How does IGFBP-4 contribute to senescence? Following endogenous or exogenous stress, can IGFBP-4 reinforce senescence by the autocrine mechanism and/or may IGFBP-4 promote senescence in healthy cells surrounding a senescent cell by a paracrine signaling?

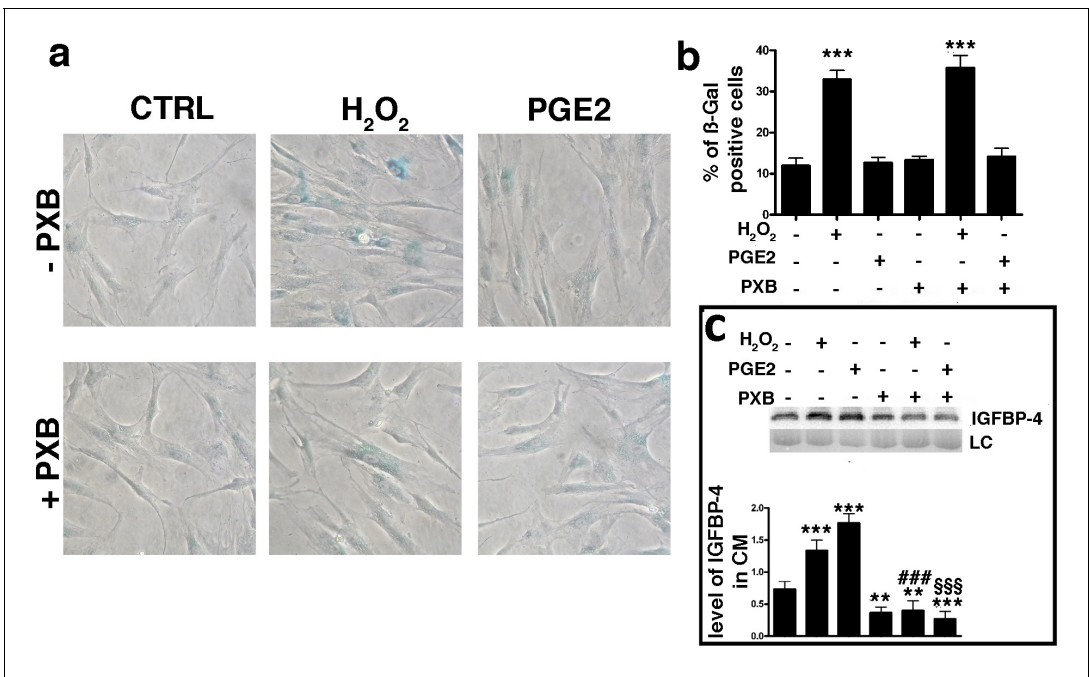

**Figure 3.** PGE2 effect on senescence and IGBP4 release. Panel **a** – Acid beta galactosidase assay in control, $H_2O_2$-treated and PGE2 treated cells. The experiments were carried out with or without 200 μM Parecoxib (PXB). The data were obtained 48 hr after genotoxic stress. The picture shows representative samples. Panel **b** – The graph shows the mean percentage of the senescent cells compared with the control ( ± SD, n = 3). Anova analysis and post-hoc test identified several statistical differences among the samples. In the picture, we indicated only differences between the untreated cells (first column) and the other experimental conditions (***$p<0.001$). Panel **c** – Western blot analysis of IGFBP-4 in the conditioned media of MSC cultures 48 hr following treatments. Experiments were carried out either in the presence of $H_2O_2$ or PXB or PGE2. Membrane staining with Ponceau S acid red was used loading control (LC). The graph shows the densitometric analysis of the IGFBP-4 level. The differences between untreated cells (first column) and the other experimental conditions are indicated with ** ($p<0.01$) or *** ($p<0.001$). The symbol ### indicates statistical differences ($p<0.001$) between $H_2O_2$ (second column) and $H_2O_2$/PXB (fifth column) treated cells. The symbol §§§ indicates statistical differences ($p<0.001$) between PGE2 (third column) and PGE2/PXB (sixth column) treated cells. The data are expressed as arbitrary units ( ± SD, n = 3).

The online version of this article includes the following source data for figure 3:

**Source data 1.** Original image data (gels and micrographs) for *Figure 3*.
**Source data 2.** Numerical data for *Figure 3*.

## PGE2-Gαs-PKA signaling is involved in IGFBP-4 secretion

An increase in the production and release of reactive oxygen species (ROS) is a typical feature of senescent cells (*Passos et al., 2010*; *Shao et al., 2011*). ROS promote the production/release of prostaglandins, which are involved in several pathological functions, such as inflammation and organ dysfunctions (*Kaneko et al., 2011*; *Miller, 2006*; *Wang et al., 2004*). Senescence and SASP production are strictly associated with these two phenomena (*Rea et al., 2018*). Based on this premise, we evaluated the level of PGE2, the most ubiquitous prostaglandin, in the medium of MSC cultures following the induction of senescence-inducing stress. We found a significant increase in the PGE2 level in the medium of MSC cultures soon after incubation with $H_2O_2$ (*Figure 1—figure supplement 1e*). We then questioned whether PGE2 production may be associated with IGFBP-4 secretion. We treated the MSCs with $H_2O_2$ in the presence of 200 µM Parecoxib, an anti-Cyclooxygenase-2 (COX-

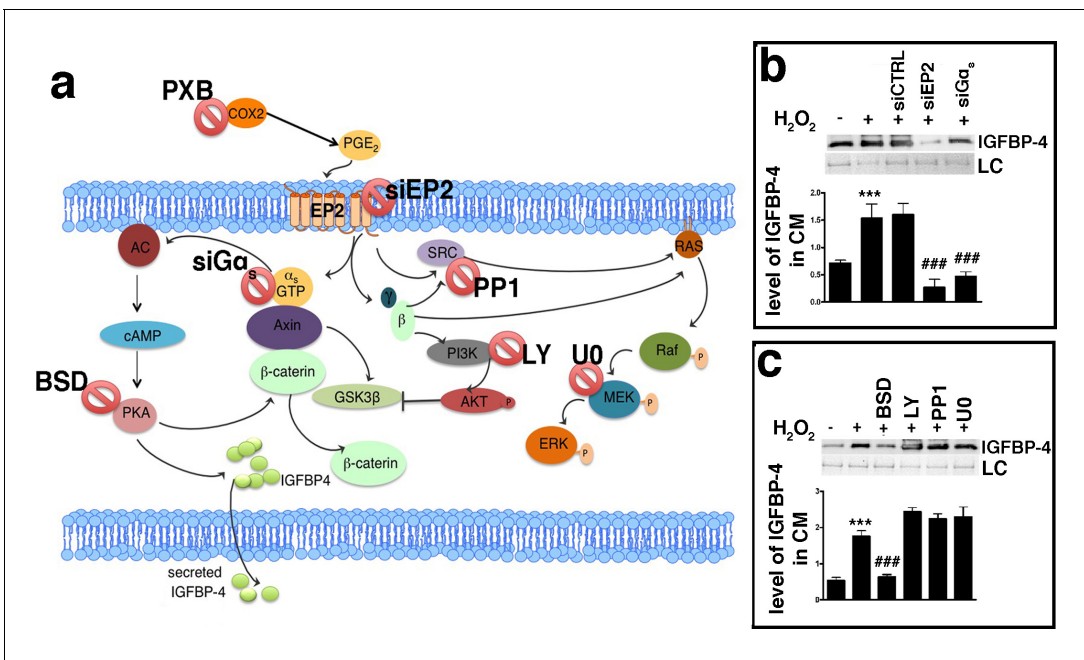

**Figure 4.** PGE2 signaling pathways. Panel **a** - PGE2 can bind the EP2 receptor. This stimulates Gαs, which activates adenylyl cyclase (AC), which produces cyclic AMP (cAMP). This nucleotide induces protein kinase A (PKA) activation. The PGE2 binding to the EP2 receptor determines the release of Gβγ subunits, which stimulate AKT through phosphatidylinositol 3-kinase (PI3K). EP2 can also promote the activation of the RAS-RAF-MEK pathway through the SRC protein. In the cartoon are depicted the drugs and inhibitors of some key factors belonging to the PGE2 signaling pathways. siEP2: siRNA against EP2 mRNA; siGαs: siRNA against Gαs; PP1: SRC specific inhibitor; LY294002 (LY): PI3K inhibitor; bisindolylmaleimide IX (BSD): PKA inhibitor; U0126 (U0): MEK1/2 specific inhibitor. COX2: cyclooxygenase 2; PXB: parecoxib. Panel **b** – The IGFBP-4 release following EP2 or Gαs silencing. The picture shows the IGFBP-4 levels in the MSC-conditioned medium 48 hr following $H_2O_2$ treatment in the presence of either siRNA against EP2 (siEP2) or against Gαs (siGαs). Control siRNAs were indicated as siCTRL. Membrane staining with Ponceau S acid red was used loading control (LC). The graph shows the densitometric analysis of the IGFBP-4 level. The data are expressed in arbitrary units. The statistical difference (p<0.001) between untreated cells (first column) with sample treated with $H_2O_2$ (second column) is indicated with *** symbol. Statistical differences (p<0.001) among $H_2O_2$ treated cells (second column) with those treated with different siRNAs (third, fourth, fifth column) are indicated with ### symbol. Panel **c** – The picture shows the IGFBP-4 levels in MSC conditioned medium 48 hr following $H_2O_2$ treatment in presence of either Bisindolylmaleimide IX (BSD) (PKA ⊥), or LY294002 (LY) (PI3K ⊥), or PP1 (SRC ⊥) or U0126 (U0) (MEK1/2 ⊥). Membrane staining with Ponceau S acid red was used loading control (LC). The graph shows the densitometric analysis of the IGFBP-4 level. The data are expressed in arbitrary units. The statistical difference (p<0.001) between untreated cells (first column) with sample treated with $H_2O_2$ (second column) is indicated with *** symbol. The statistical differences (p<0.001) among $H_2O_2$ treated cells (second column) with those treated with different drugs (from third to sixth column) are indicated with ### symbol.

The online version of this article includes the following source data and figure supplement(s) for figure 4:

**Source data 1.** Original image data (gels and micrographs) for *Figure 4*.
**Source data 2.** Numerical data for *Figure 4*.
**Figure supplement 1.** Expression level of EP2 receptors and functional tests to evaluate the effectiveness of drugs inhibiting the signaling pathway related to the IGFBP-4 release.
**Figure supplement 1—source data 1.** Original image data (gels and micrographs) for *Figure 4—figure supplement 1*.
**Figure supplement 1—source data 2.** Numerical data for *Figure 4—figure supplement 1*.

2) drug, which inhibits PGE2 production (*Padi et al., 2004*). In this condition, the cells entered senescence, but we did not detect an increase in IGFBP-4 secretion (*Figure 3a,b,c*). Cells treated with 10 nM PGE2 showed an increase of IGFBP-4 secretion 48 hr following the treatment (*Figure 3c*), but we did not observe evidence of an increase in senescence. Collectively, these results may suggest that PGE2 promotes the release of IGFBP-4 and that this protein is dispensable for the onset of senescence on stress injured cells (primary senescence); rather, it can promote the senescence on non-injured cells (secondary senescence) by a paracrine mechanism. Indeed, in a previous finding, we incubated healthy MSCs with recombinant IGFBP-4 and observed senescence onset (*Severino et al., 2013*). We evaluated which PGE2 signaling pathway is involved in the secretion of IGFBP-4 (*Dorsam and Gutkind, 2007*; *Figure 4a*). There are four different receptors for PGE2: EP1-4 ([*Sugimoto and Narumiya, 2007*]. RT-PCR analysis showed that, in MSCs, EP2 had the highest expression, while EP3 was absent (*Figure 4—figure supplement 1a*). We focused our interest on the EP2 receptor and its downstream effectors (*Figure 4a*). We induced senescence by $H_2O_2$ treatment and determines if IGBP4 secretion decreased after EP2 or Gαs silencing (*Figure 4—figure supplement 1b,c*). EP2 or Gαs downregulation strongly decreased the IGFBP-4 secretion (*Figure 4b*). The activated EP2 receptor determines the release of Gβγ subunits, which stimulate AKT through phosphatidylinositol 3-kinase (PI3K). EP2 can also promote the activation of the RAS-RAF-MEK pathway through the SRC protein. We curtailed the enzymatic activity of either SRC or MEK1/2 or PI3K by specific inhibitors (*Figure 4—figure supplement 1d,e,f*). This did not modify the IGFBP-4 release after the incubation of MSCs with $H_2O_2$ (*Figure 4c*). We then reduced the PKA activity by Bisindolyl-maleimide IX treatment (*Figure 4—figure supplement 1g*) and found that the senescence-related IGFBP-4 secretion was significantly impaired (*Figure 4c*). Globally, our finding indicates that PGE2 induces the IGFBP-4 release following its binding to the EP2 receptor and through the activation of the Gαs, PKA pathway.

## Senescence is induced through the IGFBP-4/IGF-II/IGF-IIR pathway

We then aimed to assess how IGFBP-4 could induce senescence. Insulin-like growth factors 1 (IGF-I) and 2 (IGF-II) are members of the insulin family of growth-promoting peptides. IGF-I, IGF-II, and insulin are among the most ubiquitous and abundant factors released in blood circulation (*Baxter, 2014*). IGFBP-4 and the other members of the IGFBP family modulate the activity of IGF-I and IGF-II by functioning as blood carrier proteins and regulating their interaction with insulin-like growth factor receptors 1 (IGF-IR) and 2 (IGF-IIR) (*Baxter, 2014*; *Harris and Westwood, 2012*). IGF-I interacts almost exclusively with IGF-IR, whereas IGF-II can bind either IGF-IR or IGF-IIR, with a higher affinity for the latter (*Harris and Westwood, 2012*). IGF-I and IGF-II are secreted by MSCs and then can be targets of IGFBP-4. We evaluated if IGF-I or IGF-II and their cognate receptors might take part in senescence and what the role of IGFBP-4 could be. In order to clarify this issue, we washed MSC cultures with PBS to eliminate secreted IGFs and then added to the culture medium recombinant IGF-I or IGF-II, either in the presence or absence of neutralizing antibodies against IGF-IR and IGF-IIR (*Figure 5a*; *Figure 4—figure supplement 1h,i*). The supplemental IGF-I in the culture medium did not modify the percentage of senescent MSCs, while the presence of IGF-II induced senescence (*Figure 5a*). This occurred through IGF-IIR, since blocking IGF-IIR with neutralizing antibodies abolished the IGF-II effect. The block of IGF-IR further potentiated the pro-senescence effect of IGF-II, probably by shifting all of the available IGF-II on IGF-IIR (*Figure 5a*). The copresence of IGF-II and IGFBP-4 in the culture medium induced a higher level of senescence compared to the conditions in which the medium was supplemented with only one of them (*Figure 5b*). We confirmed that the cells treated with IGF-II, whether or not in the presence of IGFBP-4, entered senescence by evaluating several markers associated with this phenomenon (*Figure 5c* through n). Cell cycle analysis showed a significant reduction of proliferating cells in the presence of IGF-II and/or IGFBP-4 (*Figure 5c*). These results agree with the reduction of Ki-67-positive cells (*Figure 5f,h*). During senescence, a significant modification of chromatin status with an increase of the heterochromatin foci may occur. Indeed, in MSCs incubated with IGF-II and/or IGFBP-4, we detected an accumulation of the HP1 foci (*Figure 5i,j*) and a reduction of H1.2 (*Figure 5f,g*). On the contrary, we did not detect any modification of macro-H2A.1 (*Figure 5i,k*). Senescence induced by IGF-II/IGFBP-4 is not due to genotoxic insult. Actually, we did not evidence the turn on of DNA repair system (ATM activation) and increase in DNA damage foci (phosphorylation of H2AX) (*Figure 5l,m,n*). In a previous paper, we showed that the senescence of human MSCs is driven mainly by RB2/P130, P53, and P27kip1



**Figure 5.** IGF-II and IGFP4 effects on senescence. Panel **a** - The histogram shows the mean percentage value of the senescent cells 48 hr following the addition of 25 ng/ml IGF-I or IGF-II to the culture medium, in the presence or absence of 2 µg/ml anti-IGF-IR or anti-IGF-IIR. The data are expressed ± SD, n = 3. We compared the untreated cells (first column) with all the other experimental conditions and statistical differences are indicated with ***p<0.001. Panel **b** – The graph depicts the cell senescent level in the MSC cultures 48 hr after the media supplementation with 25 ng/ml IGF-II and/or 35 ng/ml IGFBP-4. The data are expressed as ± SD, n = 3. We compared the untreated cells (first column) with all the other experimental conditions and statistical differences are indicated with *p<0.05 or ***p<0.001. Panel **c** - The picture shows representative FACS analysis of the MSCs grown in the presence of IGF-II and/or IGFBP-4. The experiments were conducted in triplicate for each condition. The percentages of different cell populations (G₁, S, and G₂/M) are indicated. The data are expressed with standard deviation (n = 3). We compared the untreated cells (CTRL) with all the other experimental conditions and the statistical differences are indicated with *p<0.05 or **p<0.01 or ***p<0.001. Panel **d** – Western blot analysis of senescence-related proteins in MSC treated with IGF-II and/or IGFBP-4. GAPDH expression was used as the loading control. Panel **e** - The graph shows the densitometric analysis of the proteins depicted in panel **d**. We compared the untreated cells (CTRL) with all of the other experimental conditions, and the statistical differences are indicated with *p<0.05 or **p<0.01 or ***p<0. The data are expressed in arbitrary units. Panel **f** - Representative microscopic field of H1.2 (green) and Ki-67 (red) in MSC cultures. The nuclei were counterstained with DAPI (blue). Panel **g** – The graph displays the pixel intensity of anti-H1.2 immunostaining per ROI (region of interest). Each dot corresponds to the intensity detected in a single cell. The red bar represents the mean intensity value in the different experimental conditions. We compared the untreated cells (IGF-II-/IGFBP-4-) with all the other experimental conditions and the statistical difference are indicated with **p<0.01 or ***p<0.001. The data are expressed in arbitrary units. Panel **h** - The histogram shows the percentage of Ki-67(+) cycling cells in the MSC cultures in different experimental conditions. We compared the untreated cells (IGF-II-/IGFBP-4-) with all of the other experimental conditions, and the statistical differences are indicated with *p<0.05 or ***p<0.001. Panel **i** - Representative microscopic field of HP1 (green) and MacroH2A (red) in the MSC cultures. The nuclei were counterstained with DAPI (blue). Panel **j** – The graph displays the pixel intensity of anti-HP1 immunostaining per ROI (region of interest). Each dot corresponds to the intensity detected in a single cell. The red bar represents the mean intensity value. We compared the untreated cells (IGF-II-/IGFBP-4-) with all of the other experimental

*Figure 5 continued on next page*

*Figure 5 continued*

conditions, and the statistical differences are indicated with **p<0.01. The data are expressed in arbitrary units. Panel k – The graph displays the pixel intensity of anti-Macro H2A immunostaining per ROI (region of interest). Each dot corresponds to the intensity detected in a single cell. The red bar represents the mean intensity value. The data are expressed in arbitrary units. Panel l - Representative microscopic field of γH2AX (green) and ATM (red) in the MSC cultures. The nuclei were counterstained with DAPI (blue). Panel m - The graph shows the degree of H2AX phosphorylation (γH2AX). This was evaluated by counting the number of γH2AX immunofluorescent foci per cell. The foci number was determined for 200 cells. Each dot represents an individual cell. The red bars indicate the mean value for each category (n = 3). Panel n - The histogram shows the percentage of ATM(+) cells in the MSC cultures in different experimental conditions. The data are expressed with standard deviation, n = 3.

The online version of this article includes the following source data for figure 5:

**Source data 1.** Original image data (gels and micrographs) for *Figure 5*.
**Source data 2.** Numerical data for *Figure 5*.

signaling (*Alessio et al., 2017a*). Also, in the current study, the MSC senescence induced by IGF-II and/or IGFBP-4 evidenced the activation of the same pathways (*Figure 5d,e*).

## IGFBP-4 delays IGF-II degradation

Then we questioned what could be the effect of IGF-II/IGFBP-4 interaction. Several authors demonstrated that IGFBPs can bind IGF-II and modulate its half-life (*Baxter, 2014*). We incubated IGF-II in a DMEM medium supplemented with FBS either in the presence or absence of IGFBP-4. The serum contains several proteases that can degrade IGF-II, as it can occur in vivo for circulating IGF-II (*Baxter, 2014*). We evidenced a decrease of IGF-II concentration following incubation in the culture medium with serum (*Figure 6a*). The IGF-II degradation is delayed in the presence of IGFBP-4 (*Figure 6a*). The IGFBP binding to IGF-II protects IGF-II from degradation, but impairs IGF-II interaction with its cognate receptors (*Baxter, 2014*). How could IGF-II induce senescence in the presence of IGFBP-4? There are several findings showing that IGFBPs modulate the action of IGFs, both by preventing their degradation and allowing their controlled interaction with cognate receptors. Proteolysis of IGFPBs by proteases decreases their affinity for the IGFs that are released and can interact

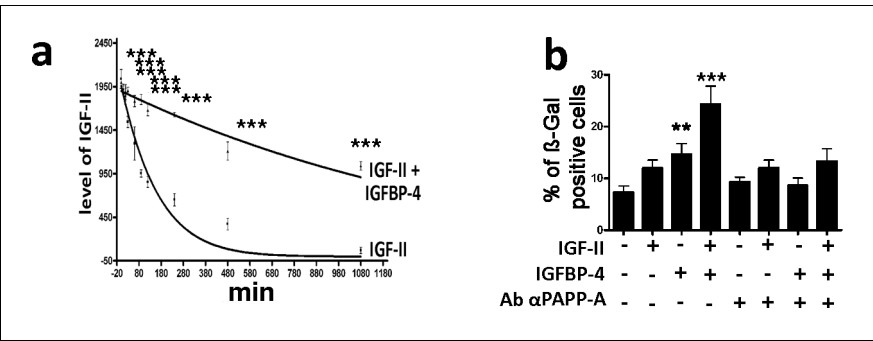

**Figure 6.** IGFBP-4 delays IGF-II degradation and modulates its pro-senescence activity. Panel a - We incubated 25 ng/ml of IGF-II in DMEM supplemented with 10% FBS for several time points. The graph shows the western blot analysis of the IGF-II levels following 10, 20, 30, 60, 90, 120, 240, 480, and 1080 min incubation either in the presence or absence of 35 ng/ml of IGFBP-4. The data are expressed as arbitrary units ± SD, n = 3. In the graph, the best fit curves are shown. For each time point, we compared the IGF-II levels in the samples either containing or not containing IGFBP-4. For each couple of corresponding samples, the statistical differences are indicated with ***p<0.001. Panel b – The graph depicts the cell senescent level in the MSC cultures 48 hr after the media supplementation with 25 ng/ml IGF-II and/or 35 ng/ml IGFBP-4. In some experimental conditions, we also added 2 μg/ml of anti-PAPP-A neutralizing antibody. The data are expressed as ± SD, n = 3. We compared the untreated cells (first column) with those incubated with IGF-II and/or IGFBP-4 (second, third and fourth column). The statistical differences are indicated with *p<0.05 or ***p<0.001. Each treatment was compared with the corresponding experimental condition in the presence of anti-PAPP-A neutralizing antibodies. For each couple of corresponding samples, the statistical differences are indicated with # p<0.05 or ## p<0.01.

The online version of this article includes the following source data for figure 6:

**Source data 1.** Numerical data for *Figure 6*.

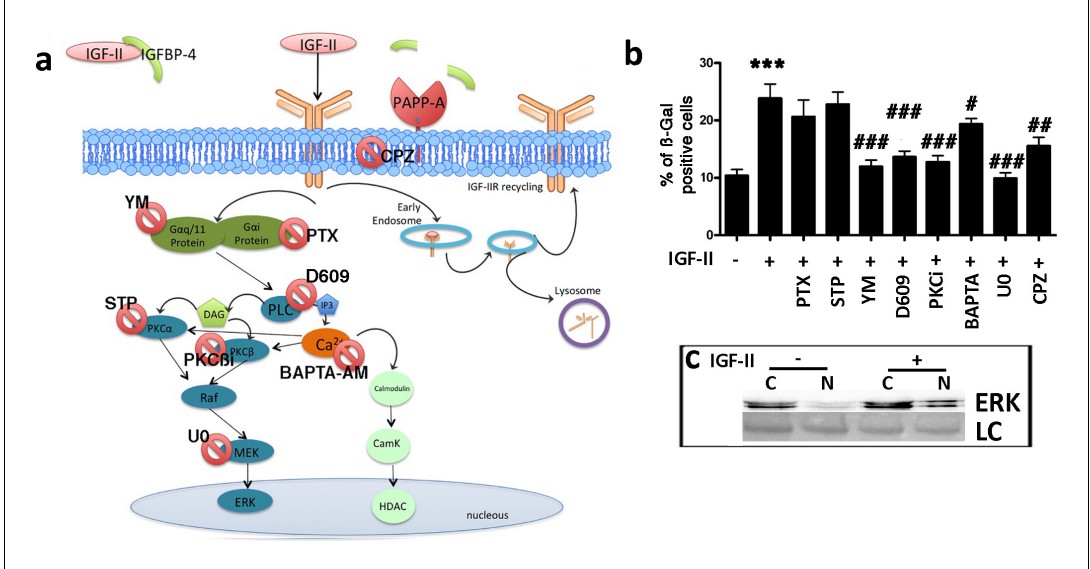

**Figure 7.** IGF-II signaling in senescence. Panel **a** – Following the degradation of IGFBP-4 by PAPP-A protease, IGF-II can bind IGF-IIR. The IGF-II/IGF-IIR complex could be internalized via clathrin-coated pits and delivered to endosomes for redistribution of IGF-IIR to the cell surface (recycling). Endosomes may also release their cargo into lysosomes for degradation. The IGF-II/IGF-IIR complex can activate an alternative pathway that is associated with the G protein and phospholipase C (PLC). The result of the PLC activity is the production of diacylglycerol (DAG) and inositol triphosphate (IP3), which in turn, can activate protein kinase C (PKC) and the RAF/MEK/ERK pathway. BAPTA-AM: 1,2-bis-(2-aminophenoxy) ethane-N, N,N',N'-tetraacetic acid tetra (acetoxymethyl) ester; CPZ: chlorpromazine; D609: tricyclodecan-9-yl-xanthogenate; PKCβi: anilino monoindolylmaleimide; PTX: pertussis toxin; STP: 1,2,3,4-tetrahydrostaurosporine; YM: YM-254890. Panel **b** - Graph depicts the cell senescent level in the MSC cultures 48 hr after the media supplementation with 25 ng/ml IGF-II with or without several drugs. The data are expressed as ± SD, n = 3. We compared the untreated cells (first column) with those incubated with IGF-II (second column) and statistical differences is indicated with ***p<0.01. The statistical differences among IGF-II treated cells (second column) with those treated with different drugs (from third to tenth column) are indicated with the # (p<0.05) or ## (p<0.01) or ### (p<0.001) symbols. Panel **c** – Western blot analysis of ERK levels in cytoplasmic (C) and nuclear fractions (N) of cells treated or not treated with IGF-II. Membrane staining with Ponceau S acid red was used as a loading control (LC).

The online version of this article includes the following source data and figure supplement(s) for figure 7:

**Source data 1.** Original image data (gels and micrographs) for *Figure 7*.
**Source data 2.** Numerical data for *Figure 7*.
**Figure supplement 1.** Functional tests to evaluate the effectiveness of drugs inhibiting the signaling pathway related to IGF-II.
**Figure supplement 1—source data 1.** Original image data (gels and micrographs) for *Figure 7—figure supplement 1*.
**Figure supplement 1—source data 2.** Numerical data for *Figure 7—figure supplement 1*.

with IGF receptors. On cell surface is present an IGFBP-4 specific protease (PAPP-A) that has a role in regulating the IGFBP-4/IGF interaction. The senescence of MSCs induced by the copresence of IGF-II and IGFBP-4 in the culture medium is significantly reduced in the presence of PAPP-A neutralizing antibodies (*Figure 6b*; *Figure 4—figure supplement 1j*).

## IGF-II induces senescence through the IGF-IIR/PLC-β/PKC-β/MEK/ERK pathway

Activated IGF-IIR determines the $G_q$ activation, which stimulates phospholipase C (PLC), which in turn, activates protein kinase C (PKC) and promotes calcium release from the endoplasmic reticulum (ER). IGF-IIR can also promote the activation of the $G_{i/0}$ pathway (*Figure 7a*; *Fernandez and Torres-Alemán, 2012*; *Hawkes et al., 2006*; *Hawkes and Kar, 2004*; *Maeng et al., 2009*). We incubated MSCs with IGF-II and, at same time, inhibited Gαi by specific inhibitor (Pertussis Toxin – PTX). IGF-II-induced senescence was not affected by blocking Gαi activity (*Figure 7a,b*; *Figure 7—figure supplement 1a*). In contrast, Gα$_{q/11}$ or PLC-β inhibition reduced the IGF-II induced senescence of MSCs (*Figure 7a,b*; *Figure 7—figure supplement 1b,c*). PKCα, PKC-βI, PKC-βII, and cytoplasmic calcium are downstream effectors of PLC-β. The inhibition of PKCα did not affect the IGF-II induced senescence (*Figure 7a,b*; *Figure 7—figure supplement 1e*). The inhibition of both PKC-β isoforms showed a reduction in IGF-II-induced senescence (*Figure 7a,b*; *Figure 7—figure supplement 1d*).

The blocking of cytoplasmic calcium with BAPTA-AM partially reduced the senescence of MSCs after incubation with IGF-II (*Figure 7a,b*; *Figure 7—figure supplement 1f*). PKC-β isoforms may activate RAF/MEK/ERK signaling. We then inhibited MEK1/2 by the U0126 drug (*Figure 4—figure supplement 1e*). In this condition, IGF-II did not induce the senescence of MSCs (*Figure 7a,b*). We confirmed that the IGF-II treatment activated the RAF/MEK/ERK signal by determining the nuclear translocation of the phosphorylated ERK1/2 (*Figure 7c*; *Figure 7—figure supplement 1h*). We then analyzed the role of ligand/receptor internalization in IGF-II-induced senescence. IGF-II and its cognate receptor, IGF-IIR, may be internalized via endocytosis. The block of endocytic process through chlorpromazine reduced the MSC senescence induced by IGF-II incubation (*Figure 7a,b*; *Figure 7—figure supplement 1g*).

## Discussion

IGFs and IGFBPs are part of a biological mechanism that has been conserved during the evolution and that Cynthia Kenyon defined as a 'conserved regulatory system for aging' (*Kenyon, 2001*). Several findings showed that some IGFBPs may be involved in the senescence process (*Baege et al., 2004*; *Sanada et al., 2018*; *Severino et al., 2013*). Some researches addressed the role of IGFBP-4 in senescence, but only a few of them tried to dissect how this factor may be related to cell genotoxic stress. Moreover, while there are studies on autocrine/paracrine SASP effects, a limited number of findings investigated possible long-distance effect of SASP components, such as IGFBP proteins. We evaluated whether genotoxic injury may promote an IGFBP-4 release in vivo, having demonstrated in vitro that senescence induced by stressful insults is associated with IGFBP-4 secretion. This factor appears dispensable for the onset of senescence on stress-injured cells, while it alone, or in combination with other SASP factors, can promote senescence on non-injured cells. We irradiated mice with a low dose of X ray and collected sera 6, 24, 48, 72, and 144 hr postirradiation. We observed a statistically significant increase of IGFBP-4 compared with untreated animals. We then questioned what the effect of a prolonged increase of circulating IGFBP-4 could be? We intraperitoneally injected the mice with IGFBP-4 twice a week for two months and then evaluated whether this increase in IGFBP-4 might be associated with the senescence phenomena. We found a significant increase of senescent cells in the lungs, heart, and kidneys. We also isolated MSCs from the IGFBP-4-treated animals and the control animals. MSCs from the IGFBP-4 group showed a significant increase in senescence compared with the controls, and this was related to a reduced stemness, as demonstrated by the CFU assay.

Altogether, these data may suggest that IGFBP-4 could act as a genotoxic stress mediator that, entering into the blood flow after its secretion by senescent cells, could promote further senescence phenomena in non-injured cells.

The limit of our study is due to technical constraints. Only a few studies investigated the secretome in vivo. Some findings analyzed the proteomes of body fluids in healthy and disease conditions but it was not possible to identify the relative contribution of a specific tissue to the body fluid proteome (*Brown et al., 2012*; *Chenau et al., 2009*; *Xue et al., 2008*). In our study it is also not possible to relate the increase of circulating IGFBP-4 to a given cell type. Nevertheless, the in vitro investigations showing that following low dose irradiation cells became senescent and secrete IGFBP-4 give credit to the hypothesis that modification of IGFBP-4 serum levels following genotoxic stress may be associated to induction of senescence phenomena.

Another interesting aspect of our study is the evaluation of IGFBP-4 in the sera of patients following CT scan. The increase of circulating IGFP-4 after low dose irradiation further confirms that this protein could be related to cellular stress. This result could contribute to better evaluate possible negative outputs associated with medical use of radiations, including the procedures that employ low dose of radiation, such as CT, scintigraphy, SPECT, (Single Photon Emission Computed Tomography) and PET (Positron Emission Tomography). Indeed, radiation exposure for medical imaging has significantly increased in the last years and our study could further support a note of caution about their improper use.

Having demonstrated a role IGFBP-4 in senescence we focused our attention on the signaling pathways associated with IGFBP-4 release and its pro-senescence activity. Genotoxic injuries are associated with ROS production, which in turn, promotes the production/release of prostaglandins. These molecules are mediators of inflammation and organ dysfunction, two phenomena strictly

associated with cellular senescence. Indeed, we found a significant increase in the PGE2 level in the media of MSCs after genotoxic stress treatment. PGE2 interacting with its receptor, EP2, promotes IGFBP-4 secretion. PGE2/EP2 induces the release of IGFBP-4 through the Gαs/PKA pathway. Our data are in good agreement with other findings showing that PGE2 may induce cellular senescence (*Dagouassat et al., 2013*; *Yang et al., 2011*). We then tried to dissect the IGFBP-4 signaling pathway that is associated with senescence. Many findings evidenced that IGFBP proteins work mainly by binding either IGF-I or IGF-II and modulating their action. In this scenario, we evaluated the function of IGF-I and IGF-II in the senescence of MSCs and the role that could be exerted by IGFBP-4. The addition of IGF-I to the MSC cultures did not affect the senescence process, while IGF-II supplementation induced senescence that occurred through IGF-IIR. The block of IGF-IR further increased the IGF-II-induced senescence, probably by shifting all of the available IGF-II on IGF-IIR. Of interest, the addition of both IGF-II and IGFBP-4 to the culture medium produced a higher level of senescence compared to conditions in which the medium was supplemented with only one of them. This increase is related to IGFBP-4 activity in delaying IGF-II degradation and allowing its controlled interaction with cognate receptors. IGFBP-4 protects IGF-II from degradation, but at the same time, blocks its interaction with cell surface receptors. Indeed, the IGFBP-4 affinity for IGFs is greater (an order of magnitude) than that of the receptors. The binding of IGF-II to its receptor can be obtained by the proteolytic degradation of IGFBP-4, which is induced by the PAPP-A protease that is present on the cell surface. According to current models, PAPP-A cleaves IGFBP-4 only when IGFs are bound to IGFBP-4. IGFs are then released close to the cell membrane, thus, facilitating receptor binding (*Conover, 2012*; *Mohan and Baylink, 2002*). We then demonstrated that senescence induced in MSC cultures, following IGF-II/IGF-IIR interaction, depends on the activation of the MEK/ERK pathway. Specifically, the Gα$_{q/11}$ proteins associated with IGF-IIR trigger the activation of PLC-β and of its downstream effectors: PKC-βI, PKC-βII. These proteins activated the RAF/MEK/ERK signaling that culminated with the nuclear translocation of phosphorylated ERK1/2. There are several reports showing that ERK1/2-related pathways may either promote or impair senescence; it depends on the cellular context, since these proteins are part of signaling hubs into cells (*Zou et al., 2019*). The phosphorylated ERK1/2 can activate a variety of substrates, including transcription factors (*Liu et al., 2018*). It could be of great interest to evaluate if IGF-II/IGF-IIR, through RAF/MEK/ERK signaling, could activate some transcription factors involved in the regulation of master regulator genes associated with senescence, such as those belonging to the P53-P21 and RB-P16 pathways. Indeed, there are some findings showing that the transcription factor ELK1, a target of ERK1/2, could play a role in senescence (*Christoffersen et al., 2010*). Of note, Boros and coworkers performed a comprehensive CHIP-chip analysis to identify the ELK1 target gene network. They identified many ELK1-binding regions, representing 1112 potential promoters. In that list, we found that the promoters of TP53 and RB2/P130 are potential targets of ELK1 (*Boros et al., 2009*). This observation is of great interest, since in a previous study, we demonstrated that P53 and RB2/P130 are the main regulators of senescence in human MSCs (*Alessio et al., 2017a*). Another interesting aspect we analyzed was the potential role of IGF-II/IGF-IIR internalization through the endocytosis pathway. Some authors suggested that the binding of IGF-II to IGF-IIR is a way to reduce the IGF-II signaling that occurs mainly through IGF-II/IGF-IR. According to this hypothesis, IGF-II/IGF-IIR undergoes endocytosis and then degradation into lysosomes. Indeed, our data suggested that IGF-II signaling may take place via IGF-IIR and that the endocytosis process of IGF-II/IGF-IIR is part of the receptor recycling process and not only a degradation pathway (*Braulke et al., 1990*). Indeed, blocking endocytosis abrogated the IGF-II pro-senescence effect.

## Conclusion

Senescence has been extensively studied in vitro, but we still require an in-depth molecular description of the in vivo properties of senescent cells. This study will allow a better understanding of the aging process and, eventually, the development of effective senolytic drugs that could eliminate senescent cells that, with their SASP, promote diseases of aging. There are some problems in performing such an investigation. One problem is related to the identification of reliable markers to identify in vivo senescent cells. The other regards the role and function of SASP. It is unclear if SASP may act only in a paracrine/autocrine manner or if SASP molecules systemically promote age-associated diseases by entering the circulation (*van Deursen, 2019*). Our finding may suggest that some

SASP components, such as IGFBP-4, may enter circulation and promote senescence in cell compartments that are far from the site of genotoxic injury.

## Materials and methods

### Human MSC cultures

Bone marrow aspirate samples were obtained from healthy donors (age 10–18 years) after informed consent. We separated cells on a Ficoll density gradient and the mononuclear cell fraction was collected We seeded $1–2.5 \times 10^5$ cells/cm$^2$ in alpha-MEM containing 10% FBS and 3 ng/mL βFGF. We used the minimal criteria suggested by the International Society for Cellular Therapy (*Dominici et al., 2006*) to identify mesenchymal stromal cells (MSC).

### Acute and chronic induction of senescent MSCs

We used three different stressors to induce acute senescence, as we already described (*Capasso et al., 2015*). Briefly: the MSC cultures at passage three were incubated either with 300 μM peroxide hydrogen (H$_2$O$_2$) (Sigma-Aldrich MO, USA) for 30 min, or with 1 μM of doxorubicin (DOXO) (Sigma-Aldrich MO, USA) for 24 hr, or they were irradiated with two different X ray doses of 40mGy (IRL) and 2000mGy (IRH) by a Mevatron machine (Siemens, Italy) operating at 6 MeV. Following each treatment, the cells were further cultivated for 48 hr. Chronic senescent MSCs were obtained by extensive in vitro cultivation for 30 days (replicative senescence, REP).

### In situ senescence-associated acid beta galactosidase assay

Cells grown in flasks were fixed using a solution of 2% formaldehyde and 0.2% glutaraldehyde for 5 min at RT. After that, the cells were incubated with a staining solution containing 1 mg/mL of X-Gal (GoldBio, MO, USA) at 37°C overnight. The percentage of senescent cells was calculated by the number of blue, β-galactosidase-positive cells out of at least 500 cells in different microscope fields, as already reported (*Zanichelli et al., 2012*).

### Cell cycle analysis and cell proliferation

For each analysis, $5 \times 10^4$ cells were collected and dissolved in a hypotonic buffer containing propidium iodide (Sigma-Aldrich MO, USA). The samples were acquired on a Guava EasyCyte flow cytometer (Merck Millipore MA, USA) and analyzed using EasyCyte software.

Cell proliferation was determined by Cell Counting Kit-8 (CCK-8) colorimetric assay for the determination of cell viability (Dojindo Molecular Technologies, Kumamoto, Japan). We seeded 5,000 cells in 96-wells and CCK-8 reagents were added. The viability was detected by a microplate reader at 450 nm 24 hr, 48 hr and 72 hr after the incubation.

### Immunocytochemistry (ICC)

We grew cells on cover slides, and then we fixed them in 4% formaldehyde solution for 15 min at room temperature. We used the following primary antibodies: HP1α (2616) and γ-H2AX (2577) were obtained from Cell Signaling (MA, USA); Ki-67 (sc7846) from SantaCruz Biotechology (CA, USA); while MacroH2A1 (ab37264), H1.2 (ab4086), and ATM (ab36810) were obtained from ABCAM (UK). All of the antibodies were used according to the manufacturer's instructions. The secondary antibodies (FITC or TRITC conjugated) were obtained from ImmunoReagents (NC, USA). Nuclear staining was performed by a DAPI mounting medium (ab104139, ABCAM, UK), and micrographs were taken under a fluorescence microscope (Leica, Germany). The percentage of ATM-, gamma-H2AX-, and Ki-67-cells was calculated by counting at least 500 cells in different microscope fields. The mean pixel intensity for H1.2-, HP1-, and Macro H2A1-cells was quantified using Quantity One 1-D analysis software (Bio-Rad, CA, USA).

### Conditioned media (CM) preparation for western blot (WB) analysis

After 48 hr of acute senescence induction or 30 days in vitro cultivation (replicative senescence), we evaluated the release of IGFBP-4 in CM. For CM preparation, the cultures were extensively washed with PBS 1x and transferred to a chemically defined, serum-free culture medium for an overnight incubation. Then, the CM were collected for WB analysis.

## Western Blot (WB) analysis

Cells were lysed in a buffer containing 0.1% Triton (Bio-Rad, CA, USA) for 30 min in ice. 20 μg of each lysate was electrophoresed in a polyacrylamide gel and electroblotted onto a nitrocellulose membrane. We used the following primary antibodies: RB1 (AV33212) and GAPDH (G8795) were from Sigma-Aldrich (MO, USA), RB2/P130 (R27020) was from BD Biosciences (CA, USA), p27$^{KIP1}$ (3686) was from Cell Signaling (MA, USA), while p107 (sc-318), p53 (sc-126), and p21$^{CIP1}$ (sc-397) were obtained from Santa Cruz Biotechnology (CA, USA), and p16$^{INK4A}$ (ab54210) was from ABCAM (Cambridge, UK). Immunoreactive signals were detected with a horseradish peroxidase-conjugated secondary antibody (ImmunoReagents, NC, USA) and reacted with ECL plus reagent (Merck Milli-pore, MA, USA). All of the antibodies were used according to the manufacturer's instructions. The mean value was quantified densitometrically using Quantity One 1-D analysis software (Bio-Rad, CA, USA).

## Nuclear cytoplasmic cell fractionation

The MSC were lysed in a buffer with 0.5% Nonidet P40. Following an incubation of 5 min at 4°C, nuclear and cytosolic fractions were obtained after centrifugation for 5 min at 500 g. The nuclear and cytoplasmic fractions were washed three times with lysis buffer and then resuspended in the lysis buffer and further centrifuged at 17,500 g. Pellets were then used for WB analysis.

## Silencing of EP2 and gαs with siRNAs

We obtained validated siRNAs targeting human EP2 (sc-40171) or Gαs (sc-29328) mRNA, as well as control siRNAs (sc-37007) from Santa Cruz Biotechnology (TX, USA). The control siRNA had a sequence that did not target any known mammalian genes. We incubated human MSCs with 100 pMoles of siRNA against EP2 (siEP2) or Gαs (siGαs) and control siRNA (siCTRL) with Lipofectamine 3000 (Invitrogen, CA, USA) for 6 hr. Then, we changed the medium and further cultivated cells for 48 hr. We evaluated the down-regulation of target mRNA by semi-quantitative RT-PCR.

## Semi-quantitative RT-PCR

Total RNA was extracted from cell cultures using Omnizol (Euroclone, Italy). The mRNA levels were measured by RT-PCR and by 5X ALL-IN-ONE RT MASTERMIX (ABM, Canada), and the Real-time PCR assays were performed with BrightGreen 2X qPCR MasterMix (ABM, Canada) and run on a Line-Gene 9600 (Bioer Technology, China). All of the reagents were used according to the manufacturer's instructions. We designed primer pairs for RT-PCR reactions with Primer Express software (Applied Biosystems, Italy) and used the mRNA sequences as templates from the Nucleotide DataBank (National Center for Biotechnology Information, MD, USA) to design primer pairs. Primer sequences are listed in *Supplementary file 3*. We used the 2-ΔΔCT method as a relative quantification strategy for quantitative real-time PCR data analysis.

## PGE2 elisa assay

We measured the level of PGE2 in a media of MSC cultures following treatment with $H_2O_2$ by a PGE2 Elisa kit (Elabscience, TX, USA) according to the manufacturer's instructions.

## PGE2 treatment of MSC cultures

MSC cultures at P3 were treated with 10 nM of PGE2 (Tocris Bioscience, UK) with or without 200 μM of Parecoxib (Dynastat, Pharmacia Europe, UK) for 24 hr at 37°C (*Li et al., 2017*).

## PGE2 pathways involved in IGFBP-4 release

We blocked several kinases associated with different PGE2 signalling circuits (*Figure 2A*) to evaluate the pathway involved in the release of IGFBP-4 following PGE2 treatment. Specifically, the MSC cultures were incubated at 37°C for 30 min with each of the following drugs, separately: 1 μM PP1 to inhibit SRC kinase; 5 μM LY294002 (LY) for the PI3K kinase; 1 μM bisindolylmaleimide IX (BSD) to block the PKA kinase, and 1 μM U0126 (U0) for MEK1/2 inhibition. Subsequently, we added 20 nM PGE2 (Tocris Bioscience, UK), and samples were further incubated for 24 hr. Finally, we harvested CM as described above. All of the inhibitors were from ProteinKinase.de (Germany). For each drug, we evaluated the inhibitory effect at the concentration we used (*Figure 4—figure supplement 1*).

### Effect of IGFBP-4, IGF-I, IGF-II on MSC senescence

35 ng/ml IGFBP-4 (code 350-05B), 25 ng/ml IGF-I (code 100–1), and 25 ng/ml IGF-II (code 100–12) were added either separately or in combination to the MSC (P3) cultures. All of the factors were obtained from PeproTech (UK). The cells were incubated for 48 hr at 37°C, and then we carried out several assays as reported in the results. In some experiments, the effects of IGFBP-4, IGF-I, and IGF-II on MSC senescence were evaluated in presence neutralizing antibodies against IGF-IR or IGF-IIR or PAPP-A.

### IGF-II half-life

We evaluated the half-life of IGF-II in the culture medium containing FBS. We incubated 250 ng of IGF-II in 1 ml of alpha-Mem with 10% FBS in the presence or absence of 350 ng of IGFBP-4 for 0, 0.5, 1, 2, 6, and 18 hr. After the incubation, samples were filtered on Microcon 10-kD (Merck Italy) to eliminate the bovine serum albumin and other high molecular weight proteins. Then the samples were treated as described in the paragraph for CM preparation.

### IGF-II signalling pathways

IGF-II signal transduction may occur through different pathways (*Figure 5a*). We determined which of these could be associated with senescence. In particular, the MSC cultures were incubated at 37°C for 30 min with each of the following drugs, separately: 100 nM Pertussis toxin (PTX, BMLG101-0050 Enzo Biochem, NY, USA) for the $G\alpha i$ kinase, 1 µM YM-254890 (YM, 10–1590 Focus Biomolecules, PA, USA) to block $G\alpha_{q/11}$, 50 µM D609 (sc201403, Santa Cruz Biotechology, CA, USA) to inhibit PLCβ, 10 µM BAPTA-AM (BPT) (15551 Cayman, MI, USA) and 10 mM 1,2,3,4-tetrahydrostaurosporine (STP, ab143861, Abcam, UK) for PKCα kinase inhibition, 5 nM PKCβi (Santa Cruz Biotech, TX, USA) for PKCβ inhibition, and 100 mM chlorpromazine (CPZ, sc-357313, Santa Cruz) to block endocytosis. Subsequently, we added 25 ng/ml IGF-II, and the samples were further incubated for 24 hr, and the senescence assays were performed. For each drug, we evaluated the inhibitory effect at the concentration we used (*Figure 7—figure supplement 1*).

### Animals

C57BL/6 inbred male mice of 3 weeks of age were purchased from Charles River (MA, USA). The animals were handled in compliance with the protocols that were approved by the Animal Care and Use Committee of Luigi Vanvitelli Campania University.

### Mice irradiation and evaluation of circulating IGFBP-4

The animals were anaesthetized with a mixture of ketamine/medetomine and then were total body irradiated with 100 mGy X Rays by a Mevatron machine (Siemens, Italy) operating at 6 MeV. At different times following irradiation, the animals were sacrificed and sera were collected for IGFBP-4 analysis. As the control, we used animals that were anaesthetized and not irradiated.

### In vivo effects of IGFBP-4 on senescence

Mice at 8 weeks of age were treated twice a week with 1 µg human IGFBP-4 for 8 weeks. The recombinant protein was intraperitoneally administered in 200 uL of PBS. The control mice were injected with 200 ul of PBS. At the end of the treatment, the animals were sacrificed by cervical dislocation, and their organs were collected for further analysis.

### Mouse MSC isolation and CFU assay

We harvested MSCs from the bone marrow of the femurs and tibias of mice. The cells from one animal were plated onto two 100 mm dishes with alpha-MEM containing 15% FBS. After 48 hr, we discarded the nonadherent cells and then incubated the remaining cells for 7 to 10 days in a proliferating medium in order to reach confluence (P0). The cells were then trypsinized and were seeded for the acid beta galactosidase assay. An aliquot of cells at P1 were used for the CFU assay. In brief, 1,000 cells were plated in 60 mm plates and were incubated for 15 days without a medium change. The plates were collected, fixed, and stained with 0.5% crystal violet. The stained colonies were identified under a light microscope and were counted.

## Histological analysis

The heart, lungs, and kidneys were harvested from mice treated with exogenous IGFBP-4 and from the corresponding controls. The organs were fixed in 2% formaldehyde and 0.2% glutaraldehyde for 1 hr at RT and were stained with acid beta-galactosidase staining solution.

Subsequently, the organs were dehydrated and embedded in paraffin, as previously described (*Rinaldi et al., 2016*). Five μm cross-sections were stained with Fast Red solution for the nucleus. Stained tissue sections were analysed and acquired by using a Leica IM1000 System (Germany). The percentage of senescent cells was calculated by the number of blue, β-galactosidase-positive cells out of at least 500 cells in different microscope fields. All of the reagents were obtained from Sigma-Aldrich (St. Louis, MO, USA) unless otherwise specified.

## Analysis of IGFBP-4 in the sera of patients following CT scan

We enrolled 10 patients that received abdominal CT scan with Siemens Somatom Definition Flash instrument. The maximum computer tomography dose index (CTDI) was 30 mGy. Inclusion criteria were: males between 18–65 years. Exclusion criteria: diabetes, obesity, severe cardiovascular diseases, cancer. We selected only male patients to reduce variability, since there are findings showing that IGF-associated pathways can be influenced by estrogen fluctuation (*Isotton et al., 2012*). Moreover, exclusion criteria are also selected to reduce variability, since the excluded conditions could alter IGFBP levels (*Baxter, 2014*; *Fischer et al., 2004*; *Renehan et al., 2004*). Patient sera were collected before CT scan and 48 hr later. Patients signed an informed consent document describing briefly the aim of the research. On these sera we determined the level of IGFBP-4 by western blot analysis. This research had the Regione Campania Ethical Committee authorization (Prot. N. 379/C. E. Campania Centro).

## Statistical analysis

Statistical significance was determined using one-way ANOVA and Tukey post hoc test by JASP software (https://jasp-stats.org). All of the statistical analyses are reported in *Supplementary file 1*. Power size calculation was determined by using CLINCAL software and STATULATOR for determining the minimum number of subjects for adequate study power (https://clincalc.com/stats/sample-size.aspx - http://statulator.com/SampleSize/ss2PM.html). Results of power size calculation are in *Supplementary file 2*.

## Acknowledgements

This work was partially supported by 2017 2018 Grants of Experimental Medicine Department (Luigi Vanvitelli Campania University) to UG, by Regione Campania Progetto POR 'Sviluppo di nanotecnologie Orientate alla Rigenerazione e Ricostruzione tissutale, Impiantologia e Sensoristica Odontoiatria/oculistica – SORRISO' CUP B23D18000250007 to GP and UG and Post-Doc Fellowship (Assegno di Ricerca Valere 2017) from Luigi Vanvitelli Campania University to NA.

## Additional information

### Funding

| Funder | Grant reference number | Author |
| --- | --- | --- |
| Regione Campania | CUP B23D18000250007 | Gianfranco Peluso<br>Umberto Galderisi |
| University of Campania Luigi Vanvitelli | 2017 2018 Grants of Experimental Medicine Department | Umberto Galderisi |
| University of Campania Luigi Vanvitelli | Post-Doc Fellowship (Assegnodi Ricerca Valere 2017) | Nicola Alessio |

The funders had no role in study design, data collection and interpretation, or the decision to submit the work for publication.

## Author contributions
Nicola Alessio, Data curation, Validation, Investigation, Methodology; Tiziana Squillaro, Conceptualization, Investigation, Methodology, Writing - original draft; Giovanni Di Bernardo, Data curation, Validation; Giovanni Galano, Roberto De Rosa, Data curation, Validation, Methodology; Mariarosa AB Melone, Supervision; Gianfranco Peluso, Supervision, Funding acquisition, Writing - review and editing; Umberto Galderisi, Conceptualization, Supervision, Funding acquisition, Writing - review and editing

## Author ORCIDs
Umberto Galderisi (iD) https://orcid.org/0000-0003-0909-7403

## Ethics
Human subjects: Bone marrow aspirate samples were obtained from healthy donors (age 10-18 years) after informed consent. Campania Region Ethical Committee approval n. 317/2016 PR.
Animal experimentation: The animals were handled in compliance with the protocols that were approved by the Animal Care and Use Committee of Luigi Vanvitelli Campania University. Italian Ministry of Health ethical approval n. 67/2012 A.

## Decision letter and Author response
Decision letter https://doi.org/10.7554/eLife.54523.sa1
Author response https://doi.org/10.7554/eLife.54523.sa2

# Additional files

## Supplementary files
• Supplementary file 1. Data from statistical analyses. For each figure the results from ANOVA and post hoc tests are provided in a separate file. We do not insert it here since it is a 40 pages document.

• Supplementary file 2. Power size calculation.

• Supplementary file 3. List of primers used for RT-PCR.

• Transparent reporting form

## Data availability
All data generated or analysed during this study are included in the manuscript and supporting files.

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
