## [Decision Letter]

**Decision letter after peer review:**

Thank you for submitting your article "Genotoxic stress induces increase of circulating IGFBP-4: implication for aging" for consideration by *eLife*. Your article has been reviewed by Anna Akhmanova as the Senior Editor, a Reviewing Editor, and two reviewers. The reviewers have opted to remain anonymous.

The reviewers have discussed the reviews with one another and the Reviewing Editor has drafted this decision to help you prepare a revised submission.

Summary:

The study aims to elucidate the relationship between genotoxic stress, IGFBP-4 protein secretion and aging. Cultured cells exposed to different stresses released IGFBP-4, which could be associated with cellular senescence; the supposed genotoxicity is evaluated by analysing in the blood of animals exposed to radiation the level of circulating IGFBP-4 and linking it to a possible pro-aging effect. The most attractive part of the paper is the comparison of these animal findings with the sera of human subjects.

Such comparative animal and human approaches are extremely interesting, even if it is not at all sure that the study will lead to approaches which will allow tackling the senescence process.

Essential revisions:

- Adapt the title as suggested by reviewer 2.

- Shorten the length of the text and avoid repetitions.

- As cellular senescence is a multifactorial process, the galactosidase test alone appears not sufficient to prove the induction of senescence, as well stressed by reviewer 1; However, I don't think it is possible to ask for more experiments before publishing this paper. The adaptation of the title and writing will allow to cope with this issue. Please also take note of the comments on Figure 5.

Please see the comments of the reviewers below for more details.

*Reviewer #1:*

The study aims to elucidate the relationship between genotoxic stresses, IGFBP-4 protein secretion, and aging. The authors supposed that IGFBP-4 released by cultivated cells exposed to different stresses is associated with cellular senescence, and evaluated whether genotoxic injury may promote an IGFBP-4 release in vivo by analyzing the level of IGFBP-4 circulating in the blood of animals exposed to radiation. In addition, they showed that the level of this protein, which is associated with cellular stress and hypothetically may be responsible of proaging effect, increased in the sera of human subjects that received Computer Tomography. Such a combination of in vitro and in vivo experiments is sorely needed in aging research, and the publication of such works will undoubtedly contribute to the further development of the field.

Besides, the authors revealed the signaling pathways associated with IGFBP-4 release and its pro-senescence activity, which expands the existing knowledge on the molecular mechanisms of aging and SASP.

However, there are some shortcomings in the work that needs to be corrected, as well as issues that need clarification and, possibly, additional research:

1) Despite the fact that the authors convincingly showed by using in vitro and in vivo models that stressful insults are associated with increased IGFBP-4 secretion, whether IGFBP-4 elevation is associated with the cellular senescence in the experiments illustrated by Figure 1—figure supplement 1, Figure 1, and Figure 3, in my opinion, remains unclear. The authors implicitly point to the induction of senescence using the galactosidase assay. Cellular senescence is a multifactorial process. The galactosidase test alone is not sufficient evidence of the induction of senescence. Additional molecular tests (expression of p21, p16, phospho-p53 and phospho-pRb) and functional assays (growth curves, monitoring of morphological changes, ROS level), similar to that which are presented in Figure 5 for IGF-related experiments, are needed. It seems especially important to prove reliably the senescence of cells (BM MSCs) obtained from the irradiated and IGFBP-treated animals in Figure 1.

2) In the context of pro-senescent effect of IGFBP-4, it would be extremely interesting to compare the level of IGFBP-4 in the medium of senescent cells and in the sera of animals exposed to irradiation or IGFBP-4, as well as in the sera of irradiated patients, using the ng/mL units obtained by ELISA.

3) Several comments on Figure 5. In my opinion, the results on the cell cycle analysis are obscure. First of all, the peaks in the histogram of the control cells are too broad (Figure 5C) in comparison to the IGF- or IGFBP-treated cells, that impedes their fitting and introduces uncertainty in determining the percentage of the S-phase cells in control samples. Anyway, the best way to compare the proliferative capacity of the control and experimental cultures is to measure the growth curves, these measurements would strengthen the author conclusions.

To sum up, basing on the results presented in this study, several mutually exclusive conclusions can be offered. In my opinion, increased IGFBP-4 secretion may be considered: (i) as a factor which is alone sufficient for cell senescence induction and/or spreading, (ii) as a factor which, in combination with other SASP factors, can promote cell senescence, (iii) as a stress mediator which is not related to the induction/spreading of senescence. It is highly recommended that authors choose and prove their point of view on IGFBP-4 pro-senescence activity.

*Reviewer #2:*

Interesting article from Galderisi et al., with the increase of IGFBP-4 in response to a genotoxic stress which could have relevance for aging. The authors assess whether the genotoxic stress may initiate the secretion in vivo in mice and humans of IGFBP-4. In mice they found an increase in senescent cells in the lung, kidneys and heart. They provide evidence on the signaling pathways elucidated by IGFBP-4. Moreover, the pathways leading to the IGFBP-4 is also described. This a comprehensive, technically sound experimental study on the role and the molecular effects of IGFBP-4 for cell senescence.

1) During the whole article the authors want to relate their findings to aging, however their experiments and results are not related to aging by any means as they only study the process of cell senescence, but this may occur at any age. They did not demonstrate that the effect of IGFBP-4 is increased or related to aging in mice or humans. The basic mechanisms elucidated by them is related to the mechanism of action of IGFBP-4 on cell senescence but not aging. The title should be modified in consequence.

2) "Acute senescence" as the author call their early (short term) genotoxic cell effects is not defined. It is not clear what does it mean in their experimental context. This is a normal physiological phenomenon which can heave a high positive action as these cells once terminated their action the immune cells may eliminate them. The chronic senescence may also have positive effects as an anti-cancer role and not meaning only negative effects. So the supposition that all these processes are "bad" should be nuanced.

3) The description of the PGE2 role and the IGFBP-secretion effects alone or together on the senescence is somehow confusing on subsection “PGE2-Gαs-PKA signaling is involved in IGFBP-4 secretion”.

4) The comment on the use of the imaging techniques in medicine qualifying them as "indiscriminate" is inappropriate in this context as the authors have no proof to make such a statement after this study which was not aimed to verify that. Moreover, the increase of IGFBP-4 after CT within 48 hours does not mean that the cell senescence increased and even less that it contributes to aging.

5) The age range of the patients for CT is very broad and the reason is not stated. They state that cancer was excluded however in many cases abdominal CT is made because a cancer is suspected.

6) This is speculative to state in the Discussion section that "prostaglandins are mediators of aging". There is no proof that prostaglandins are per se pro-aging molecules.

7) In the Discussion section there are many repetitions word by word of what has been written in the result section. These repetitions should be revised.

8) Together the article is too long.

---

## [Author Response]

Summary:The study aims to elucidate the relationship between genotoxic stress, IGFBP-4 protein secretion and aging. Cultured cells exposed to different stresses released IGFBP-4, which could be associated with cellular senescence; the supposed genotoxicity is evaluated by analysing in the blood of animals exposed to radiation the level of circulating IGFBP-4 and linking it to a possible pro-aging effect. The most attractive part of the paper is the comparison of these animal findings with the sera of human subjects.Such comparative animal and human approaches are extremely interesting, even if it is not at all sure that the study will lead to approaches which will allow tackling the senescence process.Essential revisions:- Adapt the title as suggested by reviewer 2.

We changed the title. The new one is: Increase of circulating IGFBP-4 following genotoxic stress and its implication for senescence.

- Shorten the length of the text and avoid repetitions.

We deleted two paragraphs in the Discussion section.

- As cellular senescence is a multifactorial process, the galactosidase test alone appears not sufficient to prove the induction of senescence, as well stressed by reviewer 1; However, I don't think it is possible to ask for more experiments before publishing this paper. The adaptation of the title and writing will allow to cope with this issue. Please also take note of the comments on Figure 5.

We addressed these issues, please, see comments for reviewer 1.

Reviewer #1:The study aims to elucidate the relationship between genotoxic stresses, IGFBP-4 protein secretion, and aging. The authors supposed that IGFBP-4 released by cultivated cells exposed to different stresses is associated with cellular senescence, and evaluated whether genotoxic injury may promote an IGFBP-4 release in vivo by analyzing the level of IGFBP-4 circulating in the blood of animals exposed to radiation. In addition, they showed that the level of this protein, which is associated with cellular stress and hypothetically may be responsible of proaging effect, increased in the sera of human subjects that received Computer Tomography. Such a combination of in vitro and in vivo experiments is sorely needed in aging research, and the publication of such works will undoubtedly contribute to the further development of the field.Besides, the authors revealed the signaling pathways associated with IGFBP-4 release and its pro-senescence activity, which expands the existing knowledge on the molecular mechanisms of aging and SASP.However, there are some shortcomings in the work that needs to be corrected, as well as issues that need clarification and, possibly, additional research:1) Despite the fact that the authors convincingly showed by using in vitro and in vivo models that stressful insults are associated with increased IGFBP-4 secretion, whether IGFBP-4 elevation is associated with the cellular senescence in the experiments illustrated by Figure 1—figure supplement 1, Figure 1, and Figure 3, in my opinion, remains unclear. The authors implicitly point to the induction of senescence using the galactosidase assay. Cellular senescence is a multifactorial process. The galactosidase test alone is not sufficient evidence of the induction of senescence. Additional molecular tests (expression of p21, p16, phospho-p53 and phospho-pRb) and functional assays (growth curves, monitoring of morphological changes, ROS level), similar to that which are presented in Figure 5 for IGF-related experiments, are needed. It seems especially important to prove reliably the senescence of cells (BM MSCs) obtained from the irradiated and IGFBP-treated animals in Figure 1.

You are perfectly right, β-galactosidase assay is not sufficient to claim that a cell is senescent. Anyway, we already evidenced that MSCs became senescent when treated with stress inducers, such as those shown in Figure 1—figure supplement 1 and in Figure 3. Please, see our publications:

1: Capasso et al., 2015.

2: Alessio et al.,2015.

In addition, we followed your suggestion and performed further experiments that are shown in the figure 1D,E,I and J. We modified the text as it follows:

We also isolated MSCs from mice following irradiation and observed increase senescence compared with the controls (Figure 1C). MSC senescence, we detected with acid β galactosidase assay, was confirmed by evaluating the expression of several proteins associated with this phenomenon (Figure 1E). Moreover, we observed a decline in cell proliferation (Figure 1D) along with a decrease of clonogenic potential (360 ± 36 clones per 1,000 plated cells in control vs 290 ± 23 clones in irradiated mice) (Figure 1F).

2) In the context of pro-senescent effect of IGFBP-4, it would be extremely interesting to compare the level of IGFBP-4 in the medium of senescent cells and in the sera of animals exposed to irradiation or IGFBP-4, as well as in the sera of irradiated patients, using the ng/mL units obtained by ELISA.

The increase in IGFBP-4 level in medium of MSCs following stress is indicative of what could happen in vivo following genotoxic stress. Indeed, we found an increase of circulating IGFBP-4 after irradiation both in mice and human subjects. Nevertheless, the in vitro culture is a “closed” system in which the released IGFBP-4 proteins undergo both spontaneous and proteolytic degradation. The level of circulating IGFBP-4 proteins depends not only on protein release and degradation but also on their clearance and transport into interstitial liquids (Baxter, 2014). Moreover, besides MSCs, other cell types release IGFBP-4. In this context, we believe that in vitro and in vivo comparison of IGFBP-4 concentration is poor informative. It is much more interesting the evaluation of changes in IGFBP-4 level.

3) Several comments on Figure 5. In my opinion, the results on the cell cycle analysis are obscure. First of all, the peaks in the histogram of the control cells are too broad (Figure 5C) in comparison to the IGF- or IGFBP-treated cells, that impedes their fitting and introduces uncertainty in determining the percentage of the S-phase cells in control samples. Anyway, the best way to compare the proliferative capacity of the control and experimental cultures is to measure the growth curves, these measurements would strengthen the author conclusions.

The cell cycle graph we showed in Results Figure 5C is representative of several experiments (please, seesupplementary file 1 for statistical analysis). You are right, the graph that we showed is not the best one, we changed it with another representative experiment. But please, keep in mind that the percentage of cells in the different cell cycle phases was calculated not only from data of the representative experiments but from all of them. Moreover, the reduction of proliferation following treatment of cells with IGF-II or IGFBP-4 or IGF-II + IGFBP-4 is confirmed by reduction of Ki67 staining (Figure 5H).

To sum up, basing on the results presented in this study, several mutually exclusive conclusions can be offered. In my opinion, increased IGFBP-4 secretion may be considered: (i) as a factor which is alone sufficient for cell senescence induction and/or spreading, (ii) as a factor which, in combination with other SASP factors, can promote cell senescence, (iii) as a stress mediator which is not related to the induction/spreading of senescence. It is highly recommended that authors choose and prove their point of view on IGFBP-4 pro-senescence activity.

Cells treated with peroxide hydrogen increase PGE2 levels, release IGFBP4 and undergo senescence. Cells treated with peroxide hydrogen in presence of a PGE2 inhibitor undergo senescence but do not release IGFBP4.

These results may suggest that PGE2 promotes the release of IGFBP-4 and that this protein is dispensable for the onset of primary senescence (stress induced); rather, it can promote the secondary senescence process by a paracrine mechanism. Indeed, in a previous finding, we incubated healthy MSCs with recombinant IGFBP-4 and observed senescence onset.

This statement was in subsection “PGE2-Gαs-PKA signaling is involved in IGFBP-4 secretion”.

We further clarify our point of view by adding a sentence in the Discussion section.

Reviewer #2:Interesting article from Galderisi et al., with the increase of IGFBP-4 in response to a genotoxic stress which could have relevance for aging. The authors assess whether the genotoxic stress may initiate the secretion in vivo in mice and humans of IGFBP-4. In mice they found an increase in senescent cells in the lung, kidneys and heart. They provide evidence on the signaling pathways elucidated by IGFBP-4. Moreover, the pathways leading to the IGFBP-4 is also described. This a comprehensive, technically sound experimental study on the role and the molecular effects of IGFBP-4 for cell senescence.1) During the whole article the authors want to relate their findings to aging, however their experiments and results are not related to aging by any means as they only study the process of cell senescence, but this may occur at any age. They did not demonstrate that the effect of IGFBP-4 is increased or related to aging in mice or humans. The basic mechanisms elucidated by them is related to the mechanism of action of IGFBP-4 on cell senescence but not aging. The title should be modified in consequence.

You are right, indeed we demonstrated that increase of IGFBP-4 level induces senescence and there is no direct relationship between IGFBP-4 and aging. As suggested, we changed the title in: Genotoxic stress induces increase of circulating IGFBP-4: implication for senescence.

Nevertheless, there are several papers claiming a role for cellular senescence in aging (see for example: The role of senescent cells in ageing van Deursen, 2014).

2) "Acute senescence" as the author call their early (short term) genotoxic cell effects is not defined. It is not clear what does it mean in their experimental context. This is a normal physiological phenomenon which can heave a high positive action as these cells once terminated their action the immune cells may eliminate them. The chronic senescence may also have positive effects as an anti-cancer role and not meaning only negative effects. So the supposition that all these processes are "bad" should be nuanced.

At beginning of the Results section, we better defined chronic and acute senescence, according to van Deursen, (2014). Please, see underlined text in the related manuscript. In the introduction we stated that senescence may have either positive or negative effects on health.

3) The description of the PGE2 role and the IGFBP-secretion effects alone or together on the senescence is somehow confusing on subsection “PGE2-Gαs-PKA signaling is involved in IGFBP-4 secretion”.

We tried to clarify the role of PGE2-dependent IGFBP-4 release. Please, see underlined text in the related manuscript.

4) The comment on the use of the imaging techniques in medicine qualifying them as "indiscriminate" is inappropriate in this context as the authors have no proof to make such a statement after this study which was not aimed to verify that. Moreover, the increase of IGFBP-4 after CT within 48 hours does not mean that the cell senescence increased and even less that it contributes to aging.

We changed the term “indiscriminate” in “improper”. Anyway, we would like to draw your attention on some recommendations of UNSCEAR (The United Nations Scientific Committee on the Effects of Atomic Radiation). In Germany, from 1996 to 2012, the annual effective dose per person from CT analyses has more than doubled (Schegerer et al., 2017). A recent study carried out on almost 1,000,000 adults in healthcare markets across the United States showed that a consistent number of patients received up to 50 mGy/year, a considerable value, given the reference levels for emergency provided by the International Commission on Radiation Protection (ICRP) is among 20-50 mGy/years (UNSCEAR 2010). This is outside our research but we kept in mind this study when we decided to work on patients undergoing CT scan.

5) The age range of the patients for CT is very broad and the reason is not stated. They state that cancer was excluded however in many cases abdominal CT is made because a cancer is suspected.

You are right, as stated before, we changed the manuscript title. In the Discussion section we wrote:

Altogether, these data may suggest that IGFBP-4 could act as a genotoxic stress mediator that, entering into the blood flow after its secretion by senescent cells, could promote further senescence phenomena in non-injured cells.

6) This is speculative to state the Discussion section that "prostaglandins are mediators of aging". There is no proof that prostaglandins are per se pro-aging molecules.

We are performing an in-depth study on the CT effects on human health. We aim to evaluate possible negative outputs associated with medical use of radiations. We will estimate if pro-senescence factors (IGFBP-4 and other IGFBPs) impair cellular function of tissues and organs in a patient group after CT Scan. We are enrolling a cohort of 300 patients and the data we presented in this paper refers to the first patients’ group. For this reason, the age range is broad. Anyway, according to many studies, this parameter will not affect the expected outcome or will bias results. On this premise, we did not give any detail on it. You are right, patients undergoing CT scan may have cancer or other diseases. We collected and will collect samples from a huge cohort of patients, then we will wait for diagnosis and will discard samples from patients that present some exclusion criteria, such as cancer.

7) In the Discussion section there are many repetitions word by word of what has been written in the Results section. These repetitions should be revised.

You are right, we changed the sentence in the Discussion section: “These molecules are mediators of inflammation and organ dysfunction, two phenomena strictly associated with cellular senescence.”

8) Together the article is too long.

We tried to reduce the paper and eliminated two paragraphs in the Discussion section.